# Biobased Copolyamides 56/66: Synthesis, Characterization and Crystallization Kinetics

**DOI:** 10.3390/polym14183879

**Published:** 2022-09-17

**Authors:** Chia-Hsiung Tseng, Ping-Szu Tsai

**Affiliations:** Department of Chemical and Material Engineering, College of Engineering, National Kaohsiung University of Science and Technology, Kaohsiung 807618, Taiwan

**Keywords:** biobased copolyamide, differential scanning calorimetry, eutectic behavior, nonisothermal crystallization kinetics

## Abstract

This study synthesized a series of new biobased copolyamides (co-PAs), namely PA56/PA66 with various comonomer ratios, by using in situ polycondensation. The structures, compositions, behaviors, and crystallization kinetics of the co-PAs were investigated through proton nuclear magnetic resonance (^1^H NMR) spectroscopy, gel permeation chromatography (GPC), Fourier transform infrared (FT-IR) spectroscopy, differential scanning calorimetry (DSC), polarized optical microscopy (POM), and X-ray diffraction (XRD). The influence of the composition of co-PAs on their mechanical properties and thermal stability was investigated. The co-PAs exhibited a eutectic melting point when the PA56 content was 50 mol%, with the crystallization temperature decreasing from 229 to a minimum 188 °C and the melting temperature from 253 to a minimum 218 °C. The results indicated that the tensile strength and flexural modulus first decreased and then increased as the PA66 content increased. The nonisothermal crystallization kinetics of the PA56/PA66 co-PAs were analyzed using both the Avrami equation modifications presented by Jeziorny and Mo. The results also indicated that the crystallization rate of the PA56/PA66 co-PAs was higher than that of PA56.

## 1. Introduction

Polymers derived from petroleum, natural gas, and coal have been extensively used but these conventional fossil resources are non-renewable and limited. In addition, negative externalities such as air, water, and land pollution and global warming necessitate the transition of all fossil-fuel dependent sectors to sustainable and renewable alternatives. Therefore, biobased polymers are gradually attracting considerable attention as a leading scientific and technological innovation for global economic development owing to their green, environmentally friendly, and resource-saving features that contribute to sustainable development. Polyamides (PAs) are excellent engineering thermoplastic engineering materials, which are extensively used in industrial applications. Semicrystalline PA66 is another commercialized PA that was developed by Carothers in 1936 [1] and has been extensively used in several engineering applications, including the production of membranes, films, textile fibers, electronics (e.g., connectors, surface mount devices, and reflectors), flexible packaging materials for food, and plastics for use in the automotive industry [2,3,4,5,6]. These applications are possible because of the excellent mechanical strength, dimensional stability at high temperature, chemical resistance, and other advantages afforded by PA66. PAs are becoming increasingly crucial as high-performance polymers. Several all-biobased PAs such as PA11, PA1010, and PA410, or partly biobased PAs such as PA610 and PA1012 with excellent engineering applicability, have been commercialized [7]. The biobased PAs such as PA52, PA54, PA56, PA510, and 5T (denoted as PA5X) are synthesized from 1,5-diaminopentane and diprotic acid through a green process [8,9,10,11,12,13]. 1,5-Diaminopentane is used to derive bio-PAs from renewable feedstocks to replace conventional PAs derived from petrochemical routes. Gale and Epps discovered that lysine decarboxylase from Escherichia coli can be used in acid stress-induced processes [14], thereby facilitating biobased cadaverine production. Microbial production of cadaverine has been observed in recombinant *E*. *coli* and *Corynebacterium glutamicum* strains, as indicated by plasmid- or genome-based heterologous expression of *E. coli cadA* and *ldcC* that encode inducible and constitutive lysine decarboxylase, respectively [15,16,17]. Although PA56 is biobased, it can be synthesized through the polycondensation reaction of adipic acid (AA) and 1,5-pentamethylenediamine (PMD). PA56 has been tested in the downstream sectors of the petroleum industry, not only as a renewable alternative to traditional materials, but also as a novel and unique polymer for membranes, industrial plastics, and textile markets. PA56 is characterized by high chemical resistance, melting temperature, modulus, intrinsic flame resistance, ultraviolet light stability, dyeability, and moisture absorption and can be used in applications such as industrial yarns and filaments [18,19,20,21]. However, the modification of its molecular structure has received little attention. In this work, we used PA66 to modify PA56 by in situ polymerization. The crystallization of PA56 in copolymers is usually affected not only by processing conditions but also by PA66. There is a need to design new PA56-based copolymers to reduce production costs while also maintaining or improving their mechanical, molding, thermal, and heat resistance properties.

Copolymerization methods have been extensively applied in polymer modification owing to their simplicity and efficiency. Target polymers with desirable properties can be synthesized by selecting appropriate comonomers and molecular designs [22,23,24,25]. Moreover, the use of biobased raw materials as comonomers promotes sustainable development. Considerable effort has been dedicated to preparing co-PAs from biobased comonomers [26,27,28]. Co-PAs crystallinity depends on the degree of symmetry, lengths between amide groups and orientation of amide groups (odd–even effect). In a characteristic plot of the copolymer melting temperature and enthalpy versus composition, if the melting temperatures of a copolymer do not lie at the local minimum as a function of composition, then the corresponding comonomers can be considered to be isomorphous, (e.g., the co-PA of AA and terephthalic acid in PA46/6T). PA46/6T exhibits similar lengths between the amide groups of AA and units of terephthalic acid, enabling its incorporation into the same crystal structure without inducing changes in the crystal lattice dimensions [29,30]. A local minimum indicates that the comonomer is an impurity that decreases the melting point, size, and perfection of the formed crystals and the comonomers are not co-crystalline and thus do not fit into the same crystalline lattice. Novitsky reported that neither PA10T nor PA12T is co-crystalline with PA6T. The lengths of PA10T and PA12T between the amide units are different from PA6T, and this limits the incorporation of the PA6T. A decrease in melting temperature up to 30 wt% PA6T promotes the formation of fewer and smaller or less perfect crystals [31]. However, when two crystalline polymers are copolymerized, isodimorphism occurs and is characterized by changes in crystal lattice dimensions at the eutectic point. The comonomer PA6T can be incorporated into a crystal structure that is different from its own. The degree of melting point depression and amount of crystallinity at the eutectic point are functions of the structural similarity of the repeating units of the comonomer [32,33].

In this study, a series of PA56/PA66 co-PAs with different PA56/PA66 ratio compositions were synthesized through in situ polymerization. The structure and properties of the PA56/PA66 co-PAs were comprehensively investigated using wide-angle X-ray diffraction (XRD), gel permeation chromatograph (GPC), Fourier transform infrared (FT-IR) spectroscopy, and proton nuclear magnetic resonance (^1^H-HMR) spectroscopy. Moreover, the thermal and mechanical properties of the co-PAs were systematically investigated using differential scanning calorimetry (DSC), polarized optical microscopy (POM), thermogravimetry analysis (TGA), and tensile and flexural tests. The co-PAs were subjected to practical processes, namely, molding, extrusion and film or fiber formation, under nonisothermal and dynamic conditions. Furthermore, the nonisothermal crystallization of the co-PAs was quantitatively examined in an industrial environment to optimize their processing conditions and improve their properties.

DSC is an excellent tool to measure the kinetics of nonisothermal crystalline polymerization. However, nonisothermal studies are complex and therefore require more care. In order to solve the above issues and obtain appropriate kinetic data, the Kinetics Committee of the International Confederation for Thermal Analysis and Calorimetry (ICTAC) has provided some advice and guidance [34,35]. The recommendations offer guidance for obtaining kinetic data that are adequate to the actual kinetics of various processes, including crystallization of polymers, and focus on kinetic measurements performed by means of thermal analysis methods such as thermogravimetry (TG) or thermogravimetric analysis (TGA), differential scanning calorimetry (DSC), and differential thermal analysis (DTA).

The Avrami equations modified by Jeziorny and Mo were employed to evaluate the nonisothermal crystallization kinetics of the PA56/PA66 co-PAs.

## 2. Experimental Section

### 2.1. Materials

Biobased PMD (>99.7%) was procured from Cathay Biotech Inc. (Shanghai, China). 1,6 Hexamethylenediamine (HMD; >99.7%) was procured from INVISTA (Vitoria, TX, USA). AA (>99.8%) was procured from Huafeng Group Co., Ltd., (Chongqing, China) and sodium hypophosphate (SHP) was obtained from Fuerxin Pharmaceutical and Chemical (Shangrao, Jiangxi, China). 

### 2.2. Method

#### 2.2.1. Synthesis of PA56/PA66 Co-PA Salt

The co-PA salts of PA56 and PA66 were prepared through salt reactions of PMD and HMD with individual AA samples, as illustrated in Figure 1. Six compositions of PA56/PA66 were prepared (100/0, 70/30, 50/50, 30/70, 10/90 and 0/100), and by molar ratios are presented in Table 1, with the 100/0 composition representing PA56 and the 0/100 composition representing PA66. The intermediate compositions presented random copolymers between PA56 and PA66. A 1-wt% diluted SHP aqueous solution was used as the catalyst. Weighted quantities of PMD, AA, and distilled water were added to a 10 L autoclave equipped with a stirrer, a gas inlet and an outlet. The mixture in the autoclave was stirred for 1 h at 30 °C to obtain the PA56 salt. The PA66 salt was prepared in a similar manner. After the completion of the salt preparation processes, weighted quantities (Table 1) of PA56, PA66, and SHP were added to an autoclave filled with nitrogen and agitated at 50 rpm. The co-PA salts were thus synthesized in accordance with the aforementioned approximately 1 h. 

#### 2.2.2. Synthesis of PA56/PA66 Co-PAs

The melt condensation polymerization process involved three stages: boiling of water, polymerization, and reactions for increasing viscosity. First, the autoclave was heated to 220 °C in 0.5 h and maintained at the temperature for approximately 100 min. During the heating process, the highest pressure of the autoclave was maintained at 260 psig by discharging steam. Subsequently, the pressure of the autoclave was gradually decreased to atmospheric pressure level, and the temperature was increased to 270 °C in 1 h. After the reaction, the pressure of the polymerization autoclave was decreased to −10 psig in 15 min, thereby increasing the viscosity. After the completion of the polycondensation process, the co-PAs were cooled, shaped into pellets using a pelletizing water cutter, and washed with distilled water. The co-PAs were added to a beaker and dried overnight in a vacuum oven at 80 °C. The prepared PA56/PA66 co-PAs were composed of 100, 70, 50, 30, 10, and 0 mol% PA56 and are denoted herein as PA56, CoPA-1, CoPA-2, CoPA-3, CoPA-4 and PA66, respectively.

### 2.3. Characterization

#### 2.3.1. DSC Measurements

DSC was executed using a Perkin–Elmer DSC-7 instrument (Waltham, MA, USA) to analyze the melting behavior and crystallization kinetics of the co-PAs. All DSC measurements were performed under nitrogen at a flow rate of 20 mL·min^−1^ and calibrated relative to a high-purity indium standard; samples weighing 3 and 5 mg were used to minimize the effect of low PA thermal conductivity. To determine the melting temperatures and crystallization temperatures, the co-PA samples were heated to 300 °C at a constant rate 20 °C·min^−1^ for 5 min to eliminate residual crystals; the samples were then rapidly cooled to 15 °C (at −10 °C·min^−1^) to obtain the cooling curve. In the final stage, the samples were heated to 300 °C (at 10 °C·min^−1^) to obtain the heating curve.

#### 2.3.2. FT-IR and ^1^H-NMR

The room-temperature ^1^H NMR spectra of the PA56/PA66 copolymers were obtained using a JEOL ECZ600R 600 MHz spectrometer (Tokyo, Japan). The FTIR spectra were recorded using a PerkinElmer FTIR instrument (Waltham, MA, USA). At least 32 scans were performed for each sample to obtain spectra at a spectral resolution of 2 cm^−1^ within the wavelength range of 600–4000 cm^−1^ by using a KBr disc.

#### 2.3.3. Gel Permeation Chromatography (GPC)

The number average molecular weight (*M_n_*), weight average molecular weight (*M_w_*), and polydispersity index (PDI) of the PA56/66 copolyamides were determined via the gel permeation chromatography (GPC) device from a Waters alliance system (Malvern Ltd., Malvern, UK) with refractive index (RI) and water high-resolution (HR) column at a flow rate of 0.6 mL/min and injection column of 250 μL. Column temperature was held at 35 °C. The well-characterized narrow poly(methyl methacrylate) (PMMA) in hexafluoroisopropanol (HFIP) was used as the calibration.

#### 2.3.4. TGA Measurements

TGA was performed using a TGA550 thermobalance (TA Instruments, New Castle, DE, USA). Samples weighing 9−11 mg were placed in a platinum pan in a nitrogen atmosphere and heated from 25 to 800 °C at a rate of 10 °C·min^−1^. The characteristic degradation temperature of the derivative thermogravimetric curve was determined at a 5% weight loss ($T_{d-5\%}$).

#### 2.3.5. Polarized Optical Microscopy (POM)

The spherulitic morphology of the PA56/PA66 co-PAs was observed using a Nikon LV100ND (Tokyo, Japan) polarized optical microscope equipped with a LINKAM LTS420 (Surry, UK) hot stage and a DS-Fi3 camera system. The samples were sandwiched between two microscopic slides and melted at 300 °C for 5 min and then cooled to 200 °C at −10 °C·min^−1^; the images of the samples were captured at different times and after crystallization. This process was accompanied by the appearance and growth of spherulites.

#### 2.3.6. X-ray Diffraction (XRD)

The XRD patterns of the PA56/PA66 co-PAs were obtained using an X-ray diffractometer system (Rigaku, D/MAX, Tokyo, Japan) equipped with linear monochromatic $\mathrm{Cu}–K_{\alpha }$ radiation (λ = 0.154 nm) at ambient temperature and operated at 40 kV and 200 mA. The system was operated within a 2θ range of 5–50° and at a scanning speed of 5° min^−1^.

#### 2.3.7. Mechanical Properties

Tensile strength and flexural strength tests were performed using a universal testing machine (Z010, Zwick, Ulm, Germany). The sample used for the measurements was 3.25 mm thick, 10 mm wide, and 114 mm long. The crosshead speed was set to 20 mm/min.

#### 2.3.8. Nonisothermal Crystallization Process

The nonisothermal crystallization kinetics of the PA56/PA66 co-PAs were investigated. The samples were rapidly heated from 15 to 300 °C at 80 °C·min^−1^ and then melted for 10 min at 300 °C to erase their thermal history. Subsequently, the samples were cooled to 15 °C at the five different cooling rates: 2.5, 5, 10, 20 and 40 °C·min^−1^. The exothermal curves of heat flow as a function of temperature were recorded.

## 3. Results and Discussion

### 3.1. Characterization of PA56/66

FT-IR spectroscopy was conducted to obtain detailed information about the structures of PA56/PA66 co-PAs on the basis of the characteristic vibration energies of various molecular groups. Figure 1 displays the FT-IR spectrum of PA56, PA66, and a series of PA56/PA66 co-PAs, revealing characteristics bands ascribed to aliphatic PAs. The bands at approximately 3295 and 3086 cm^−1^ were attributed to the N–H stretching vibrations in the crystalline and amorphous parts. The bands at approximately 2930 and 2858 cm^−1^ were attributed to N–H in-plane bending vibrations (symmetric and asymmetric). The peaks at approximately 1630 cm^−1^ were attributed to C–O stretching vibrations (amide I). Moreover, bands corresponding to N–H bending and C–N stretching vibrations (amide II) were observed by 1536 cm^−1^ and resembled those corresponding to the α form bonds of various nylon structure, reported by Arimoto, Bradbury and Elliott [36]. Bands were also observed at 1464 and 1417 cm^−1^ and were ascribed to deformation vibrations of CH_2_ adjacent to N–H and C=O groups, respectively. A band was also noted at 936 cm^−1^ and resembled a typical band for PA56 and PA66 macromolecules; this band was ascribed to C–C(O) stretching vibration. All FT-IR spectra indicated that the synthesized PA56/PA66 co-PAs possessed desirable properties. The differences between the spectra obtained for the PA56, PA66 and PA56/PA66 co-PAs were negligible [37,38,39,40,41].

The chemical structures of the PA56/PA66 copolymers were analyzed using ^1^H-NMR spectroscopy, as displayed in Figure 2. Signals were observed at approximately 3.3 ppm (a) and 2.4 ppm (b) and were ascribed to the methylene of the adipate unit. Moreover, signals were observed at approximately 1.9 ppm (*e + h*), 2.3 ppm (*d + g*), and 4.0 ppm (*c + f*) and corresponded to the methylene in hexanediamine and pentanediamine units. Overlapping characteristic peaks were observed, and these were attributed to the high similarity between hexanediamine adipate and pentanediamine adipate. However, the enhancement of the signal strengths of peaks (*e + h*) for all samples corresponded to an increase in hexanediamine adipate content, similar to the increase in the feed ratio of hexanediamine during polycondensation reactions [42,43,44]. The integral areas of resonance peak could be used to confirm the composition of PA56/66 copolyamides from the signal integrals of peaks at *a*, *b*, (*d + g*), and (*e + h*) using the following equations, for which the results are presented in Table 2 [44].
(1)$$b+\left(d+g\right)-a=4x+4y$$
(2)e+h=4x+2y
(3)$$PA56=\frac{y}{x+y}\times 100\%\;;PA66=\frac{x}{x+y}\times 100\%$$
where, *x* and *y* are the mole% of PA56 and PA66, respectively. The ratios of integral areas of the proton peaks in ^1^H-NMR spectra were approximately equal to the composition, which proved that the chemical structures of the PA56/PA66 co-PA match the expectation.

Molecular weight and its distribution of the synthesized PA/56/66 co-PAs were characterized by GPC. The calculated number average molecular weight, weight average molecular weight, was shown in Table 2. It can be seen clearly that molecular weight between acceptable values for polycondensation polymers; specifically, values were between 19,700–33,400 g/mol and 43,600–53,600 g/mole for *M_n_* and *M_w_*, respectively. A well relationship was found: the lower values were detected for PA56/PA66 co-PAs with an intermedia composition and higher values for neat PA56 and PA66. The *M_n_* and *M_w_* of CoPA-2 were much lower, and molecular weight distribution also known as polydispersion index (PDI) was 2.21, higher than other co-PAs. The larger the PDI, the broader the molecular weight. The main reason for this phenomenon is the co-PA becomes more irregular and the CoPA-2 is more disordering.

### 3.2. Thermal Properties of PA56/PA66

Figure 3a displays the TGA curves derived for the PA56/PA66 co-PAs as well as the temperatures corresponding to a weight loss of 5 wt% ($T_{d-5\%}$), which ranged between 298.0 and 383.5 °C. The neat PA56, PA66 and their copolyamides have slight weight loss at 100 °C, which are 0.769%, 0.661%, 0.619%, 0.813%, 0.339% and 0.555 wt%, respectively. This phenomenon due to the evaporation of free water in copolyamides, and after the temperature 100 °C, there is still continuous weight loss, which is caused by the evaporation of crystal water in the polyamide. This crystal water is the formation of hydrogen bonds between the water molecules and the amide end of the polyamide. As indicated in this figure, the obtained co-PAs possessed excellent thermal stability because the $T_{d-5\%}$ values of all PA56/PA66 co-PAs were over 300 °C except for that of CoPA-2, which was 298 °C. All curves exhibited one stage of weight loss. The $T_{d-5\%}$ value of CoPA-2 was lower than those of the other samples; the $T_{d-5\%}$ value first decreased and then increased as the mole percent of PA66 increased (Table 3). The CoPA-2 sample did not form perfect crystals at the eutectic point of the PA56/PA66 co-PAs. The values of the extrapolated onset temperature (*T_onset_*), temperature at the maximum rate of decomposition (*T_max_*), temperature at 50% weight loss (*T*_50%_), and residue at 600 °C are listed in Table 3. *T_max_* was obtained according to the DTG curves, as displayed in Figure 3b. The $T_{max}$ values of the samples ranged from 442.8 to 457.0 °C, and CoPA-2 had the lowest $T_{max}$ value. The amounts of residue of PA56 and PA66 were higher than those of all PA56/PA66 co-PAs, indicating that the homo-PAs possessed strong H bonding and more perfect crystalline structures [45,46].

### 3.3. Differential Scanning Calorimetry (DSC)

The second heating DCS curves of PA56, PA66 and PA56/66 co-PAs after cooling at DSC 10 °C·min^−1^ from the molten states are shown in Figure 4. The results of DSC are listed in Table 4. The results show that the melting behavior of PA56 is more complicated by comparison with that of PA66. As illustrated in Figure 4a, the melting curves of the CoPA-1, CoPA-2, and CoPA-3 samples exhibited double-melting endotherms, which are common for PAs, and a recrystallization phenomenon was observed [47]. The melting temperatures of PA56 and PA66 were 253 and 261 °C, respectively, and the crystallization temperatures were 208 and 229 °C, respectively. In the range of PA56 content of 0–50 mol%, the melting temperature and crystallization temperature of co-PAs showed an approximate linear decrease trend with the increase in PA66 content, and the lowest *T_m_* and *T_c_* values were observed for CoPA-2, which were 218 and 188 °C, respectively, as shown in Figure 4c. Similarly, the observed melting enthalpies ((*H_m_*), crystallization enthalpy ((*H_c_*) and glass transition temperature (*T_g_*) of co-PAs with an intermedia composition were lower compared to that of neat PA56 and PA66, the (*H_m_* (50.6 J·g^−1^), (*H_c_* (−46.9 J·g^−1^) and *T_g_* (54.2 °C) values of the CoPA-2 are lowest. The above phenomenon describes how when more PA66 units are introduced into the PA56 molecular chain, the ordered arrangement of the PA56 molecular chain is destroyed, and the co-PA structure becomes more irregular; this results in crystal defects and the formation of small crystals, reducing the crystallization ability and adversely affecting the melting temperature. At 50 mol% PA66, PA56/PA66 forms a eutectic point, and the copolymer chain structure is the most disordered. The main reason for this is that there is a significant chain length difference between PMD and HMD, and after HMD replaces PMD, the asymmetric hydrogen bonds and the conformation of the chain are broken [31,48]. When PA66 exceeds 50 mol%, the *T_m_* and *T_c_* values of co-PAs also increase with the increase in PA66 content, and the melting and crystallization behaviors of co-PAs gradually tend to the pure PA66.

### 3.4. Crystal Structure of PA56/PA66

Bunn and Garner reported that nylon 66 and nylon 610 possess triclinic structures consisting of fully extended chains connected by hydrogen bonds and that their X-ray diffractograms exhibit two characteristic peaks at room temperature [49]. Most even–even nylon materials possess α-form crystals at room temperature. The main characteristic diffraction peak of γ-crystalline PAs is a near-hexagonal unit cell with a diffraction peak at approximately 2θ = 21.0°. Crystalline forms have been observed for odd–odd, odd–even and even–odd PAs [50]. In this study, XRD (Figure 5) was used to analyze the crystallization of the PA56/PA66 co-PAs. As displayed in Figure 5, the XRD pattern of PA56 and CoPA-1 exhibited strong reflections at 2θ = 21.22° and 23.39°, respectively and weak reflections at 2θ = 21.13° and 24.17°, respectively; the strong reflections at 2θ = 21.22° and 23.39° corresponded to the (020) and (010) crystallographic planes, respectively and the weak reflections at 2θ = 21.13° and 24.17° corresponded to typical γ and α crystal forms, respectively. However, only one strong diffraction peak was observed for CoPA-2 at 2θ = 21.61°, similar to the diffraction peaks observed for the γ-crystalline phase of PA56 reported by Puiggáli and colleagues [51,52]. With an increase in PA66 units, the characteristic diffraction peaks observed for PA56/PA66 broadened, and their intensity decreased. This is because low levels of PA56 disrupted PA66 crystal formation. Beyond the eutectic point at 50 mol% PA66, the two diffraction peaks reappeared, and the WAXD patterns gradually became similar to that of PA66 with a further increase in PA66 content. The XRD spectra of CoPA-3 and CoPA-4 are displayed in Figure 5; maximum peaks were observed at 2θ = 20.38°, 23.36° and 20.74°, 23.09°, respectively, which corresponded to the (100) and (010)/(100) crystallographic planes, similar to those observed for PA66 (2θ = 20.74° and 23.42°), indicating the presence of the α-crystalline phase (Figure 5). The PA56 and PA66 segments complemented each other during the crystallization process. The WAXD patterns exhibited a characteristic peak for PA56 when the PA56 content was dominant. The regular arrangement of the amide bonds in the PA66 segments was broken, hindering the crystallization of the PA56 molecular chain. For PA66, a characteristic crystallization peak began to appear when the PA66 content was adequate to form a longer fragment.

### 3.5. Mechanical Properties 

Figure 6 illustrates the effects of PA56 on the mechanical properties of the samples. The tensile strength and flexural strengths at different PA56/PA66 ratios were determined to be within the ranges 71.6–86.7 and 117.0–136.4 MPa, respectively. The tensile/flexural strengths of neat PA56 and PA66 were 80.4/131.7 MPa and 86.7/136.4 MPa, respectively. and those of the copolymers exhibited an approximately linear decreasing trend to where the PA66 content was 0–50 mol%. When the PA66 content exceeded 50 mol%, the tensile strength and flexural strength of the co-PAs gradually increased and tended toward those of neat PA66. The molecular structure and crystal form of copolyamide directly determine the mechanical properties of the copolyamide. Compared with hexamethylenediamine, the comonomer pentamethylenediamine has one less methylene group in the molecule, and PA56 is an odd–even polyamide, and its asymmetric structure will influence the structural properties of PA56/66 co-PAs. The addition of the PA66 segment to PA56 increased the disordering of PA56, resulting in a decrease in the mechanical properties of co-PAs. When the PA56 content reaches 50%, the CoPA-2 copolyamide has the most disordered sequence structure, the lowest tensile strength and flexural strength (Figure 6). As show in Table 4, the melting temperature *T_m_* and glass transition temperature *T_g_* of the coPA-2 is lower than other co-PAs. The PDI of Co-PA2 is higher, and a wider molecular weight distribution leads to the poor mechanical property.

In addition, it can be seen from Figure 7c that the structure of CoPA-2 also presents more and smaller spherulites, resulting in more grain boundaries and surface area, so CoPA-2 is easily destroyed by external force. When the PA66 content exceeded 50 mol%, as the content of the PA66 segment increased continuously, the disordering decreased, the melting temperature *T_m_* and glass transition temperature *T_g_* increases again, the tensile strength and flexural modulus tended toward those of neat PA66.

### 3.6. Crystallite Morphology

POM is among the most predominant and effective tools for investigating spherulitic morphologies [53,54,55,56]. Figure 7 compares the polarized optical micrographs of the neat PA56, neat PA66 and co-PAs after they had been melted at 300 °C, annealed for 5 min, and then quenched to *T_c_* = 200 °C at 10 °C·min^−1^. From this figure it can be clearly seen there is a large variation in the size and morphology of spherulites in all the samples. For the neat polymer PA56, well-defined large, impinged Maltese cross spherulites were observed, as illustrated in Figure 7a. However, a dense granular texture of crystals was formed for the PA56/PA66 co-PAs, and the sizes of the crystalline grains were smaller. These results thus demonstrate that the number of crystal nuclei increased and the growth rate of crystalline grains increased with the PA66 content. The spherulites in the CoPA-2 sample, as displayed in Figure 7c, were more uniform than those in the other co-PA. The CoPA-2 sample exhibited the smallest structure. PA56, PA66, and the co-PAs were supercooled at a specific temperature to allow crystal growth in order to observe and compare their morphologies clearly during the crystallization process, as shown in Figure 8. The crystal size of CoPA-1 was smaller than that of neat PA56, indicating that a large proportion of the nucleus was generated in a short period and that the growth of the crystal was retrained.

### 3.7. Nonisothermal Crystallization Kinetics 

To study the nonisothermal crystallization kinetics of PA56/PA66 co-PAs, the relative crystallinity *X*(*t*) was determined using the following equation:(4)$$X\left(t\right)=\frac{Q_{t}}{Q_{\infty }}=\frac{\int_{T_{0}}^{T}\;\left(\frac{dH_{c}}{dt}\right)dt}{\int_{T_{0}}^{T_{\infty }}\left(\frac{dH_{c}}{dt}\right)dt}$$
where the *T**_o_* and *T*_∞_ are crystallization onset and end temperature, respectively. *T* is an arbitrary temperature and *dH_c_*/*dt* is the heat flow rate; $Q_{t}$ is the heat generated at temperature *T*; $Q_{\infty }$ is the total heat generated during the final crystallization process. Figure 9 displays the nonisothermal crystallization exothermic peaks observed for a series of the PA56/PA66 co-PAs obtained under different cooling rates *Φ*; the crystallization peak temperature *T** values, crystallization peak times $t_{max}$ and relative crystallinity levels are summarized in Table 5. It could be found that *T** shifted to a lower temperature region and became wider as the cooling rate increased from Figure 9. The relative crystallinity *X*(*t*) values obtained at different cooling rates through DSC are illustrated in Figure 10. A series of reversed *S*-shaped curves was observed, and these were attributed to spherulite impingement in the final crystallization stage; moreover, the curves tended to flatten. The crystallization time and crystallization temperature can be formulated as follows (Figure 10):(5)$$t=\frac{\left|T_{0}-T\right|}{\Phi }$$

The plots of crystallization *t* on the *x*-axis are displayed in Figure 11. All these curves exhibited similar sigmoidal profiles.

The crystallization process was determined to be highly dependent on temperature. The most common approach for describing isothermal crystallization kinetics is the classical Avrami equation [57,58,59]:(6)$$X\left(t\right)=1-\exp\left(-kt^{n}\right)$$
which is often written in a double logarithmic form as follows:(7)$$\log\left\{-\ln\left[1-X\left(t\right)\right]\right\}=\log k+n\log t$$
where *n* is the Avrami exponent, which is dependent on the type of nucleation process; *t* is the time; and *k* is the crystallization rate constant involving both nucleation and growth parameters. The crystallization half-time, $t_{1/2}$ is described as the time at which the relative degree of crystallization is equal to 50% and can be calculated from the following equation:(8)$$t_{1/2}={\left[\frac{ln2}{k}\right]}^{\frac{1}{n}}$$

The rate of crystallization G is the reciprocal of $t_{1/2}$; $G=\tau_{1/2}={\left(t_{1/2}\right)}^{-1}$. The values of $t_{1/2}$ and $\tau_{1/2}$ are listed in Table 5. The $t_{1/2}$ values decreased as the cooling rate increased for each sample. The $t_{1/2}$ values for the PA56/PA66 co-PAs were lower than that for neat PA56. The parameter $t_{max}$ indicates the time required to achieve the maximum crystallization rate. Because $t_{max}$ corresponds to the point at which $dQ\left(t\right)/dt=0,$ *Q(t)* can be defined as follows:(9)$$t_{max}={\left(\frac{n-1}{nk}\right)}^{\frac{1}{n}}$$

As shown in Figure 9, $t_{max}$ values were derived from the heat flow curves by using Equation (9). The $t_{max}$ value of the PA56 sample was higher than those of neat PA66 and PA56/PA66 co-PAs (Table 5). 

#### 3.7.1. Modified Avrami Equation by Jeziorny

Under the assumption of a constant crystallization temperature, Mandelkern [60] described that the primary stage of nonisothermal crystallization using the Avrami equation as follows:(10)$$1-X\left(t\right)=\exp\left[-Z_{t}t^{n}\right]$$
(11)$$\log\left\{-\ln\left[1-X\left(t\right)\right]\right\}=n\log t+\log Z_{t}$$
where $Z_{t}$ is a growth rate constant involving nucleation and growth rate parameters and *n* is the mechanism constant, which depends on the type of nucleation and growth process. To use the equation for analyzing nonisothermal crystallization behavior, Jeziorny [61] considered that the values of $Z_{t}$ determined by the Avrami equation should be modified as follows:(12)$$\log Z_{c}=\frac{\log Z_{t}}{\Phi }$$

On the basis of Equations (7) and (8), we created plots of $\log\left\{-\ln\left[1-X\left(t\right)\right]\right\}$ versus *log t* (Figure 12). The values of *n*, $Z_{t}$ and $Z_{c}$ were determined from the slope and intercept of the plots. For the nonisothermal crystallization process (Figure 12), the values of *n*, $Z_{t}$, and $Z_{c}$ for the PA56/PA66 co-PAs at each cooling rate are listed in Table 6. The curves derived for the PA56/PA66 co-PAs were divided into two sections representing the primary crystallization stage and secondary crystallization stage. The curves indicated the occurrence of a secondary crystallization stage in the nonisothermal crystallization processes of the PA56/PA66 co-PAs. Therefore, the fitting area was selected within the relative crystallinity range of 0–80%. During the initial primary stages of crystallization, the Avrami exponent *n_1_* varied from approximately 2.68 to 5.26 for CoPA-1, 2.74 to 3.81 for CoPA-2, 3.61 to 4.56 for CoPA-3 and 3.46 to 4.04 for CoPA-4, indicating the occurrence of a three-dimensional growth phenomenon [57]. Compared with the values of *n*_1_, those of *n*_2_ were 1.21–2.71 for CoPA-1, 1.01–2.73 for CoPA-2, 0.96–2.41 for CoPA-3, and 0.97–1.20 for CoPA-4; these values considerably decreased because of spherulite impingement in the later stages of crystallization [62,63]. The Avrami exponent *n*_2_ was approximately 1 (Table 6), and this was attributed to spherulite impingement and crowding. When the spherulites exhibited one-dimensional growth, the crystallization mode became simpler [64] during the secondary stage. The $Z_{t}$ values were strongly dependent on the cooling rate and increased with the cooling rate; this is because at higher cooling rates, crystallization occurred at lower temperatures. Compared with the values of $Z_{t}$ at the same cooling rate, the crystallization rate constant $Z_{t}$ for the PA56/PA66 copolymers was higher than that derived for the neat PA56 sample, and CoPA-2 exhibited a higher crystallization rate.

#### 3.7.2. Mo Method

To develop a more appropriate model for depicting nonisothermal crystallization, Mo [65,66] has adopted a convenient approach that entails combining the Avrami equation with the Ozawa equation. The Mo equation can be used to efficiently describe nonisothermal crystallization and successfully characterize the nonisothermal crystallization behaviors of nylon 11 [62], nylon 12 [63], nylon 66 [67], nylon 69 [68], nylon 46 [69], nylon 1212 [70] and biobased nylon composites [71,72]. Accordingly, the following equations can be derived:(13)$$\log\;Z_{t}+n\log t=\log K\left(T\right)-m\log\Phi$$
(14)$$\log\Phi =\frac{1}{m}\log\left[\frac{K\left(T\right)}{Z_{t}}\right]-\frac{n}{m}\log t$$
where the parameters $F\left(T\right)=\left[K\left(T\right)/Z_{t}\right]{\;}^{1/m}$ and *a* = *n*/m; *a* is the ratio between the Avrami and Ozawa exponents, the parameter $F\left(T\right)$ refers to the value of cooling rate choses set at a unit crystallization time at which the system has a certain degree of crystallinity. The smaller the value of $F\left(T\right)$, the higher the crystallization rate becomes. On the basis of these assumptions, the following equation can be derived:(15)$$\log\;\Phi =\log\;F\left(T\right)-a\log\;t$$

According to Equation (15), at a given degree of crystallinity of the PA56/PA66 co-PAs, plotting of log Φ versus log t (Figure 13) yields a linear relationship between log Φ and log t. Using a straight line to fit these data points, we obtained straight lines with a slope of −*a* and intercept of $\log\;F\left(T\right)$. It can be seen from Table 7 that $F\left(T\right)$ increases with the increase in the relative degree of crystallinity. The values of *a* remained almost constant during the crystallization of each sample, indicating that the model of crystallization did not change during crystallization. It can be seen that the $F\left(T\right)$ value of PA56/PA66 co-PAs (Table 7) increase with the increase in relative crystallization, and indicates that a higher cooling rate is required to quickly reach a higher relative crystallinity when they crystallize from melt. Therefore, the Mo model could accurately account for the nonisothermal crystallization kinetics of the PA56/PA66 co-PAs. By contrast, the crystallization model changed depending on the PMD/HMD ratio, signifying that the PMD/HMD ratio affected the model of crystallization of PA56/PA66 co-PAs. At a particular relative crystallinity, the $F\left(T\right)$ values derived for the PA56/PA66 co-PAs were lower than that derived for the neat nylon PA56 sample, with the value for CoPA-2 being the lowest. These results were consistent with those obtained using the Jeziorny method.

## 4. Conclusions

First, the PA56/PA66 co-PAs with various comonomer contents were successfully synthesized using in situ polycondensation. The structures of the co-PAs were analyzed through FT-IR spectroscopy and ^1^H-NMR. The crystalline structures, morphologies, melting behaviors, crystallization behaviors, and eutectic behaviors of PA56/PA66 were investigated using WAXD, POM and DSC. Both PA56 and PA66 exhibited a eutectic melting point when the PA56 content was 50 mol%, and they demonstrated the most disordered molecular structure, as well as a low melting and crystallization temperatures. In addition, the POM results indicate that the size of spherulites in PA56/PA66 was significantly small and that the number of spherulites was large; therefore, several “induced nuclei sites” were formed because of the reordering of the hydrogen bonds inside the copolymer system. Furthermore, the tensile strength and flexural modulus first decreased and then increased as the PA66 content increased. TGA revealed that the obtained co-PAs possessed excellent thermal stability. This study also investigated the nonisothermal crystallization kinetics of the PA56/PA66 co-PAs through DSC. The equations presented by Jeziorny and Mo were employed to evaluate the nonisothermal crystallization kinetics of the PA56/66 co-PAs. The results obtained using the Jeziorny equation indicated that the nonisothermal crystallization could involve a two-stage process. In the primary stage, the n_1_ value ranged from 2 to 6, corresponding to a three-dimensional and complex mechanism. In the secondary stage, the n_2_ value ranged from 1 to 3. The n_2_ value decreased relative to the n_1_ value because of spherulite impingement and crowding. The Mo equation was used to successfully describe the two stages of the nonisothermal crystallization process of PA56/PA66. The values of $F\left(T\right)$ derived for PA56/PA66 strongly depended on the crystallization rate and were lower than those derived for neat PA56 at the same relative crystallinity. The crystallization rates of the PA56/PA66 co-PAs increased relative to that of neat PA56. Therefore, The PA66 may serve as a nucleating agent for crystallization. 

## Data Availability

The data presented in this study are available on request from the corresponding author.

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
