# Peer review of "Biobased Copolyamides 56/66: Synthesis, Characterization and Crystallization Kinetics"

_polymers, 2022, doi:10.3390/polym14183879_

Round 1

Reviewer 1 Report

In this work authors presents synthesis of biobased copolyamides PA56/PA66 with various composition as possible more sustainable materials to the existing on the market polyamides. The thermal, mechanical properties and kinetic of non-isothermal crystallization of copolymers PA56/PA66 and PA56 and PA66 were investigated. Generally, the paper is well written and is interesting. However, the results presentation should be improved. Therefore, I recommend to accept the manuscript after major revision. Below are listed some issues which require consideration:

1. The molecular weight and its distribution have a effect on the thermal and physical properties of polymers and also on crystallization kinetics. The molecular weight of the synthesized copolyamides PA56/PA66 and PA66, PA56 should be determined a

2. The real composition of the synthesied copolyamides should be determined from NMR spectra. From NMR spectra the degree of randomness for the obtained copolymers should be also calculated and presented.

3. Figure 4 presents DSC heating and cooling scans for the obtained polyamides, but only Tm and Tc were analyzed. The melting and crystallization enthalpy should be also presented and discussed. The glass transition temperature is also an important parameter for polymers and should be determined and disused for the presented synthesized polyamides.

4. It is known that the applications of PA56 in some areas are frequently limited by its high moisture absorption rate and low thermal stability compared with PA66.  On page 7 in Figure 3a, the weight loss in the temperture renge between 100 and 200 oC is visible for all the obtained polyamides. What is related to this  weight loss? It is water loss?

5. On page 9  it is stated that “When the PA66 content exceeded 50 mol%, the tensile strength and flexural strength of the co-PAs gradually increased and tended toward those of neat PA66.” Authors should give possible explanation of such behavior.

Yours sincerely. 

The Reviwer

Author Response

Thanks very much for taking the time and effort to review this manuscript. I really appreciate all the comments and suggestions that have enabled us to improve our work. Based on the instructions provided in the cover letter, we uploaded the revised manuscript file. Accordingly, we have uploaded a copy of the revised manuscript with the changes highlighted by using the track changes mode in MS Word. Appended to this letter is our response to the comments raised by the reviewers. The comments are reproduced, and our responses are given afterward in a different color (red). We would also like to thank you for allowing us to resubmit a revised manuscript.

We hope that the revised manuscript is accepted for publication in the Polymers.

Sincerely

Pingszu Tsai

Reviewer 2 Report

Minor revision!   The manuscript of Chiah-Hsiung Tseng and Ping-Szu Tsai is dedicated to obtaining Copolyamides. The paper considers copolymers with different comonomer ratios. Structural, thermal and optical methods are used to characterize them. The work was done to a decent standard. On the one hand, the literary review is not very large, but at the same time it includes several dozen sources (links). The methodological part is presented in a classical form and does not raise additional questions. I would like to note that the authors took into account the recommendations of the Kinetics Committee of the international Confederation for Thermal Analysis and Calorime- try (ICTAC), which, in my opinion, is important and that some authors periodically have to pay attention to. Unfortunately, there is no line numbering in the article and therefore it is difficult to work with it. 3.1. "accribed" perhaps the authors had entered "ascribed". Figure 1. I recommend increasing the font size in the figure. Figure 3. Need to improve the quality of the figure. Figure 5. I recommend the authors to follow the format of the figures (in this case, the symbols can be placed on the figure). 3.5. mechanical properties. This size I recommend the authors to describe in more detail. The question also arises, the authors write "The above phenomenon is that when more PA66 units are introduced into the PA56 molecular chain, the ordered arrangement of the PA56 molecular chain is destroyed, and the co-PA structure becomes more irregular", how do the authors explain the increase in strength with increasing PA66 content? "The spherulites in the CoPA-2 sample, as displayed in Figure 7(c), were more uniform than those in the other co-PA" - it's hard to agree with this statement. The photo quality is not very good.

Author Response

(The authors gave the same response as above.)

Round 2

Reviewer 1 Report

All comments were included in the revised version of the article. It can be accepted in revised version to publish.